# New Insights into the Roles of lncRNAs as Modulators of Cytoskeleton Architecture and Their Implications in Cellular Homeostasis and in Tumorigenesis

**DOI:** 10.3390/ncrna8020028

**Published:** 2022-04-13

**Authors:** Carlos García-Padilla, María del Mar Muñoz-Gallardo, Estefanía Lozano-Velasco, Juan Manuel Castillo-Casas, Sheila Caño-Carrillo, Virginio García-López, Amelia Aránega, Diego Franco, Virginio García-Martínez, Carmen López-Sánchez

**Affiliations:** 1Department of Human Anatomy and Embryology, Faculty of Medicine, Institute of Molecular Pathology Biomarkers, University of Extremadura, 06006 Badajoz, Spain; evelasco@ujaen.es (E.L.-V.); garcialopez@unex.es (V.G.-L.); virginio@unex.es (V.G.-M.); 2Department of Experimental Biology, University of Jaen, 23071 Jaen, Spain; mmmg0012@red.ujaen.es (M.d.M.M.-G.); jmcc0028@red.ujaen.es (J.M.C.-C.); scano@ujaen.es (S.C.-C.); aaranega@ujaen.es (A.A.); dfranco@ujaen.es (D.F.); 3Fundación Medina, 18016 Granada, Spain

**Keywords:** lncRNAs, actin filaments, cytoskeleton architecture, cancer, tumorigenesis

## Abstract

The importance of the cytoskeleton not only in cell architecture but also as a pivotal element in the transduction of signals that mediate multiple biological processes has recently been highlighted. Broadly, the cytoskeleton consists of three types of structural proteins: (1) actin filaments, involved in establishing and maintaining cell shape and movement; (2) microtubules, necessary to support the different organelles and distribution of chromosomes during cell cycle; and (3) intermediate filaments, which have a mainly structural function showing specificity for the cell type where they are expressed. Interaction between these protein structures is essential for the cytoskeletal mesh to be functional. Furthermore, the cytoskeleton is subject to intense spatio-temporal regulation mediated by the assembly and disassembly of its components. Loss of cytoskeleton homeostasis and integrity of cell focal adhesion are hallmarks of several cancer types. Recently, many reports have pointed out that lncRNAs could be critical mediators in cellular homeostasis controlling dynamic structure and stability of the network formed by cytoskeletal structures, specifically in different types of carcinomas. In this review, we summarize current information available about the roles of lncRNAs as modulators of actin dependent cytoskeleton and their impact on cancer pathogenesis. Finally, we explore other examples of cytoskeletal lncRNAs currently unrelated to tumorigenesis, to illustrate knowledge about them.

## 1. Introduction

Recently, extensive advances in transcriptomic and genomics have led to elucidate that the genome is far more pervasively transcribed than was previously appreciated, highlighting the importance of the non-coding genome in several cellular processes [1,2,3]. Broadly, the non-coding genome can be divided into short non-coding RNAs, such as microRNAs, and long non-coding RNAs. Much of the newly discovered transcriptome appears to represent long non-coding RNAs (lncRNAs), a heterogeneous group of largely (more than 200 nucleotides) uncharacterized transcripts without the potential to generate proteins [4,5]. Similar to mRNAs, lncRNAs molecules are transcribed by RNA polymerase II and in a few cases by RNA polymerase III and are subjected to post-transcriptional modifications such as the addition of a poly (A) tail at the 3′ end or a cap at the 5′ end. Structurally, they present exons and introns and undergo alternative splicing processes [6,7,8]. Evolutionary analysis has demonstrated that lncRNAs are lowly conserved across species but display a very restricted expression both in the tissue and time lapse where they are expressed [9]. Although the majority of lncRNAs are found in the nuclear genome, several examples of lncRNAs located in the mitochondrial genome have been described. Mitochondrial lncRNAs are transcribed and processed by the mitochondrial transcriptional machinery but are under the regulation of nuclear proteins [10]. At the cellular level, lncRNAs can be located both in the nucleus and in the cytoplasm, but some cases of lncRNAs located in both cell compartments have been reported. Notably, cytoplasmic lncRNAs usually act as post-transcriptional regulators while nuclear lncRNAs usually act as transcriptional regulators reflecting a correlation between location and function of lncRNA [11,12,13]. Multiple reports have demonstrated the pivotal role of lncRNAs in multiple cellular processes such as proliferation, differentiation, development, cytoskeletal remodeling, homeostasis, and diseases [14,15,16,17,18].

Classically, the cytoskeleton has been defined as a complex and dynamic meshwork composed of different structural proteins and a multitude of accessory components, mostly of a protein nature that mediates formation and stabilization of cytoskeletal structures themselves and the interactions between them. Broadly, cytoskeletal architecture is formed by three pivotal protein structures: (1) microfilaments of actin, responsible for shaping the cell and movement in a multitude of cells and unicellular organisms; (2) microtubules consisting of tubulin dimers, which facilitate the anchoring of the different cellular organelles in the cytosol as well as in the movement of the genetic material during cell cycle progression; and (3) intermediate filaments formed by proteins of different natures with a purely structural function (Figure 1). The interactions between these three protein structures allow a single cell to be able (1) to adapt its shape in response to both chemical and physical-mechanical signals received from the environment, particularly the extracellular medium; (2) to establish contact with surrounding cells, constituting a cell monolayer; (3) to move through different tissues; and (4) to promote genetic exchange during the cell cycle [19,20]. 

Cytoskeletal homeostasis is under intense and tight regulation at both transcriptional and post-transcriptional levels, triggering cytoskeleton remodeling and turnover of cytoskeletal structures. Transcriptional regulation involves activation and inactivation of conserved gene pathways such as RHO GTPases /Rho or Wnt/β-catenin signaling, while post-transcriptional regulation modulates the processing of several genes of these pathways [21,22]. In most cases, loss of cytoskeletal homeostasis leads to pro-environmental disease as a consequence of the mis-regulation of cell cycle progression, the location of cell organelles, cell–cell junction integrity, increased ROS and ER stress levels, and disestablishment of the vesicular traffic and cell shape. As result, many pathological processes are triggered. For example, several reports have highlighted dysregulation of cytoskeleton architecture as a pivotal hallmark in cancer [23]. Loss of cell–cell junctions by cytoskeletal alterations is required for malignant cells to be able to metastasize and induce migration and invasiveness of many tumours, enhancing their aggressiveness. On the other hand, mis-regulation of actin associated proteins is related to increased ROS levels and ER stress which aids the progression and survival of several tumours. Furthermore, inhibition of cytoskeletal remodeling results in a lower ratio of proliferation and growth of carcinomas, pinpointing it as a possible therapeutic target [24,25]. Emerging evidence has pointed out the importance of lncRNAs as modulators of cytoskeletal remodeling involved in tumorigenesis by several mechanisms, such as settling proteins of cytoskeleton, modulating their expression, and/or regulating gene signal cascades involved in actin dependent structures [26,27]. Indeed, the expression of most cytoskeletal lncRNAs described to date displays upregulation in several cancer types and is correlated with enhanced aggressiveness and poor prognoses in patients (Table 1). In this review, we summarize actin-cytoskeletal lncRNAs involved in tumorigenesis, highlighting their importance as oncogenes in migration and metastasis development. Furthermore, the role of lncRNAs during the modulation of the cytoskeleton is not exclusively found in cancer, as these are also implicated in other diseases such as atrial fibrillation (AF), as detailed here.

## 2. lncRNAs as Cytoskeletal Modulators of Cellular Homeostasis

Alteration of cytoskeletal architecture is involved in the genesis of several diseases such as cardiovascular diseases (heart failure or atrial fibrillation), neurodegeneration related diseases (Alzheimer’s disease), and chronic liver diseases (steatohepatitis, copper toxicosis, or cholestasis and in different tumors) [42,43,44,45]. Many reports have related several diseases to the dysregulation of cytoskeletal modulator pathways such as Wnt/B-catenin, RHO GTPases ases/ROCK1-2, ROS system, and UPRs [46,47,48,49,50]. Recently, we have described in our lab several unknown lncRNAs involved in atrial fibrillation, a highly prevalent arrhythmogenic cardiac disease. A biotinylated pull-down assay of Walras followed by mass spectrometry analysis demonstrated interactions between several cytoskeletal proteins such as MYH9, ACTN4, TLN-1, or RhoA. Validation analysis confirmed physical binding between Walras and ACTN4 suggesting a possible cytoskeletal role. Functional knockdown of Walras by siRNA translated into reduced atrial cardiomyocyte cell size and impaired actin fiber distribution that resulted in loss of cell–cell contact, demonstrating a pivotal role of Walras in cytoskeletal homeostasis [28] (Figure 2A).

Another example of cytoskeletal lncRNA was provided by Chen et al. (2017). They demonstrated a pivotal role of taurine upregulated gene 1 (TUG1) in cortex cytoskeleton formation of vascular smooth muscle cells (VSMCs). TUG-1 is required for the correct assembly of EZH2 and the α-actin complex, participating on it. Downregulation of TUG1 has revealed that impaired expression of TUG1 triggers the breakdown of EZH2-α-actin interaction, leading to accelerated depolymerization of F-actin in VSMCs. Curiously, EZH2 mediated methyltransferase action upon α-actin is necessary for correct lysine-methylation, and thus the formation of the cortex cytoskeleton [51]. Functional assays have proven that TUG1 is capable of physically binding to both EZH2 and α-actin, modulating EZH2 methyltransferase activity. Furthermore, methylation of α-actin was inhibited by knockdown of TUG1. In conclusion, these findings suggested that EZH2-mediated methylation of α-actin may be dependent on TUG1 and thereby promotes cortex F-actin polymerization in VSMCs [52] (Figure 2B).

## 3. lncRNAs Modulators of Actin Filaments and Accessory Proteins in Cancer Pathogenesis

Actin microfilaments are considered the most dynamic of the three cytoskeletal protein structures capable of strong structural changes within minutes, determining the shape of a cell [29]. Under physiological conditions, actin is present in two different configurations: as a free globular monomeric protein, G-actin, and as a long semi-flexible helical filament, F-actin, formed by the polymerization and assembly of a multitude of protein units of G-actin. Originally, the configuration of F-actin requires the formation of dimers or trimers of G-actin, a process known as cytoskeleton nucleation that requires the participation of accessory cytoplasmic proteins such as Arp2/3 factor. Later, F-actin emerges from primed G-actin dimers or trimers by addition of free cytosol monomeric G-actin in an ATP-dependent mechanism. Cytosolic G-actin is bound to ADP. In order to polymerize and form F-actin, phosphorylation of ADP to ATP is required, which is hydrolyzed when it is incorporated into the emerging filament [53,54]. Curiously, G-actin is polarized, and therefore, F-actin is polarized as well, with the less dynamic side termed as the negative-end (−) and highly dynamic positive-end (+) results in a ten times higher polymerization rate than the (−)—end. Polymeration of G-actin is an ATPase dependent process, (+)—and (−)—ends can also be distinguished by their ATP/ADP status, being (+)—end, which contains higher amounts of ATP bound actin while the (−)—end contains more ADP bound actin [55]. Modulation of assembly and disassembly of actin filaments is one of the first phenotypic effects displayed by cancer cells. Cytoskeletal reorganization allows malignant cells to move through tissues and colonize new tissue niches leading to metastasis and a considerable increase in tumor aggressiveness [56,57]. Furthermore, tumor growth requires intense cell division that is translated into a higher proliferation rate. Interestingly, actin filaments are also required for cell cycle progression forming the cytokinesis ring that separates the original cell into two daughter cells [58]. Cytoskeleton homeostasis and remodeling requires a multitude of protein factors such as Formin like 1 (FMNL1) and Arp2/3 (polymerization factors), Cofilin (CFL-1) and Profilin (PFN1) (regulators of ADP-G-actin addition to forming filament), F-actin and α-actin (cross linker factors), or Cdc42, RhoA, Rac1, and RHOCK (signaling molecules involved into cytoskeletal remodeling) [59]. Recently, the impact of the non-coding genome in cancer related actin-cytoskeleton architecture has been demonstrated. As a consequence, many reports have pointed out the relevance of microRNAs and lncRNAs as modulators of actin filaments promoting or inhibiting the proliferation, migration, and invasion of malignant cells in several carcinomas (Figure 2) [16,60].

An example of the most direct interaction of lncRNA-actin filament in a cancer context has been provided by Pei et al. (2018) [61], which described a new lncRNA, named LNC CRYBG3, involved in the formation of the cytokinesis ring during cell cycle division. Deep sequencing techniques during cancer-radiation approach therapies showed upregulation of LNC CRYBG3 in several lung cancer cell lines, suggesting that it is a radiosensitive induced lncRNA. Functional assays demonstrated that LNC CRYBG3 binds directly to monomeric G-actin preventing it from polymerizing into F-actin (Figure 3A). Inhibition of F-actin polymerization by LNC CRYBG3 inhibits the formation of contractile ring during cytokinesis, and therein incomplete cytoplasmic division is observed. Expression analysis of LNC CRYBG3 during the cell cycle showed a correlation with the different cell cycle stages, exhibiting highest expression in G0/G1 phase and lowest in M phase. Curiously, F/G actin ratio is lowest in the G0/G1 phase and highest in both the G2/M and M phases. As result of impaired cellular cytokinesis, over-expressed LNC CRYBG3 lung cancer cell lines display an arrest cell cycle in G2/M phase, resulting in increased apoptosis and reduced cellular proliferation, migration, and invasion, pointing out the importance of LNC CRYBG3 as a tumor suppressor lncRNA. In order to explore the functional impact in different radiosensitive therapeutic approaches, in vivo assays should be performed [61].

LNC CRYBG3 is not the only example of lncRNA involvement in cell cycle progression by modulation of cytoskeletal actin dynamics. Grembergem et al. (2016) showed that breast cancer cell lines require upregulation of a long non-coding RNA cytoskeleton regulator RNA (CYTOR), previously annotated as LINC000152, to increase cellular proliferation and to promote the transition between the G2/M phase in malignant cells. Functional assays have demonstrated that downregulation of CYTOR leads to the repression of several proteins involved in cytoskeletal homeostasis such as Golph3, Rhobtb3, and plakophilin4, which in turn leads to the failure of actin filament remodeling. Furthermore, Golph3 is essential to cytoskeletal molecular pathway activation such as in mTOR while PKP4 is recognized as a structural protein in cell–cell and cell–matrix binding [62,63]. As a result of an impaired cytoskeletal configuration, CYTOR knockout cells exhibit a reduction in both their cell size and cellular periphery along with a lower ratio of proliferation and increased apoptosis. Curiously, the upregulated expression of CYTOR has been reported not only in vitro models but also in patients with several breast cancer classes such as HER2 or triple negative carcinomas subtypes [64]. Moreover, emerging evidence has pointed out that CYTOR is not a breast cancer specific lncRNA, showing upregulated expression in several other types of cancer such as nasopharyngeal, gastric, colon, or hepatocellular carcinoma and also correlating with poor prognoses and increased aggressiveness [30,31,65,66,67,68].

Cytoskeletal dynamics require the participation of several modulating factors which regulate G-actin assembly and de-assembly to form filaments. Among the modulators reported, those in the actin depolymerizing factor (ADF)/cofilin protein family are essential for actin dynamics, cell division, chemotaxis, and tumor metastasis. CFL-1 is a primary non-muscle isoform of the ADF/cofilin protein family accelerating the actin filament turnover in vitro and in vivo [69]. Interestingly, expression of Cofilin-1 is upregulated in early steps of several carcinomas. Loss-of function assays performed in different cancer cell lines have demonstrated the importance of this protein in promoting proliferation, invasion, and migration of malignant cells; however, it does not seem to have an effect on tumor size [70,71,72]. Functional assays have led to clarify that defects in cell proliferation are a consequence of G1 phase arrest during cell cycle while reduction in metastasis potential is a response to lamepodia disruption. Since the pivotal role of Cofilin-1 as an oncogene, the search for a molecular pathway modulating its expression has become a priority [73,74].

Recently, two lncRNAs have been identified as modulators of Cofilin-1 expression/function in different types of gliomas–Growth Arrest-Specific 5 (Gas5) and Aldoa repressor specify transcript (ARST) [75,76]. Curiously, the expressions of both are downregulated in glioma tissues of different grades and glioma cell lines. Zhao et al. (2015) [75] demonstrated that Gas5 repressed miR-222 expression in glioma cell lines, acting as a competitive endogenous lncRNA (ceRNA) (Figure 3B). Of note, miR-222 is a well-known inducer of proliferation and migration in several carcinomas, including glioma [32,77,78]. Binding of Gas5-miR-222 results in promoting expression of two pivotal protective factors, Bcl-2 modifying factor (Bmf) and Plexin C1 (PLXN1), which are direct targets of miR-222. While Bmf is required for the expression of several apoptotic genes such as Bcl-2 or Bax, Plexin 1 is a key repressor of Cofilin-1 activation by modulating its phosphorylation state [79,80,81]. As a consequence of Bcl2 and Bax upregulation and an increased intracellular inactive cofilin-1 pool, Gas5 over-expression in U87 and U251 glioma cell lines is translated into a lower ratio of proliferation and migration and enhanced cellular apoptosis, suggesting that Gas5 can negatively modulate glioma aggressiveness by altering the miR-222/Bmf/PLXN1 axis [75].

Unlike Gas5, ARST does not affect Cofilin-1 mRNA and/or protein expression levels but modulates functional binding of Cofilin-1 to F-actin (Figure 3C). Gain and loss of function assays demonstrated that ARST mediates actin fibers integrity by directly interacting with ALDOA protein preventing it from attaching to F-actin binding sites, which in turn are occupied by Cofilin-1. As a result, dynamic balance between polymerization, enhanced by ALDOA and depolymerization exerted by cofilin-1, is disrupted, displaying a higher rate of F-actin disassembly. In support of the altered balance of cytoskeleton remodeling, ARST overexpressed U87 and U251 glioma cell lines exhibit obvious morphological changes losing dendrite-like shape by disruption of actin stress fibers. Furthermore, ARST upregulation leads to the inhibition of malignant phenotypes of glioma cells, reducing cellular proliferation and tumor size and enhancing cellular apoptosis [76]. Taking into account results described above both Gas5 and ARST can be defined as tumor suppressive lncRNAs, suggesting a possible therapeutic use of them to target cofilin-1 expression and/or function in gliomas.

Cells are usually surrounded by an extracellular matrix (ECM), and adhesion of the cells to the ECM is a pivotal step in their migration process [33,34,82,83]. Integrins are important receptors for the ECM and together with other proteins such as talin, paxillin, or focal adhesion kinase constitute cytoskeletal structure assembly terms as focal adhesions (FAs). Stabilizing this type of contact between cells and the extracellular matrix requires the participation of actin filaments which are linked to FA through several filamentous proteins, such as vinculin or Tensin1 (TSN1) [84]. Formation and disassembly of FAs is dynamically regulated during cell migration. Curiously, turnover of polymerization and depolymerization of FAs is increased in several carcinomas since it is required for the movements of maligned cells through different tissues leading to metastasis and enhanced tumor aggressiveness [85]. However, the molecular events underlying the dynamics of FA assembly are less well understood. Emerging insights have revealed a pivotal role of lncRNAs as modulators of “dynamics of FAs”, promoting metastasis of maligned cells in different tumors such as breast or hepatocellular carcinomas.

Chang et al. (2021) [86] described an unknown mouse lncRNA named Mammary Tumor Associated RNA 25 (MaTAR25), a nuclear enriched and chromatin associated lncRNA, which it is upregulated in different breast cancers including luminal, triple negative, and HER2 subtypes [86]. Mechanistically, MaTAR25 physically interacts at a nuclear level with PURB, a purine rich element binding protein, which in turn promotes the expression of several factors involved in cytoskeleton architecture [87] (Figure 3D). Curiously, binding of MaTAR25-PURB is required for proper transcription of PURB-dependent genes. Among them, Tensin1 stands out, as a critical component in focal adhesions allowing connection and contact between extracellular matrix and actin filaments. High levels of Tensin1 have been positively correlated both in cell migration and invasion of different carcinomas. Furthermore, repression of Tensin1 expression is translated into impaired cell–cell and cell–matrix linking [88,89]. Knockout MaTAR25 in 4t1 cell line results in lower viability, migration, and cellular invasion accompanied by Tensin1 downregulation, suggesting that its expression is modulated in trans by MaTAR25. As results of repression of Tensin1, MaTAR25 knockout cells exhibit a disruption of actin filaments integrity and failure of focal adhesion formation. Interestingly, subcutaneous injection with antisense oligonucleotides (ASOs) against MaTAR25 reduces tumor growth in vivo, pointing out a possible therapeutic target of breast cancer. Comparative homology analysis has identified lncRNA LINC01271 (hMaTAR25) as a human orthologue of MaTAR25. To explore whether LINC01271 and MatTAR25 shared the same oncogene function in breast cancer, functional assays in vitro were performed demonstrating that the downregulation of LINC01271 results in lower ratio of cell viability similar to that the observed in MaTAR25 downregulation, suggesting that it may play a role in human breast cancer progression, and thus, it could be considered as a promising diagnostic and/or therapeutic target [86].

Several gene pathways have been related to focal adhesion establishment, and their deregulation is reported in several tumors. Extensive sequence transcriptome analysis performed in samples of hepatocellular carcinoma have identified DARS-AS1 as oncogene positively modulating the expression of Cytoskeleton associated protein 2 (CKAP2), which in turn promotes the expression of two pivotal pathways required for remodeling cytoskeleton derived focal adhesions, focal adhesion kinase (Fak) and extracellular signal-regulated kinases (ERK) [90,91]. Functional assays have demonstrated that DARS-AS1 act as sponge-lncRNA binding to miR-3002 which exert a protective role against HHC targeting of CKAP2 3′UTR and reducing cellular proliferation and migration [92] (Figure 3E).

The dynamic movement of malignant cells through different tissues is related to the emission of cytoplasmic projections resulting in the reorganization of actin filaments based on the cell surface. Interestingly, the ratio of cellular protrusions determinates the migratory and invasive abilities of malignant cells [33]. Long non-coding RNA urothelial cancer-associated 1 (UCA1) has been reported as an upregulated oncogene factor in bladder cancer, positively modulating the formation of actin-dependent cell protrusions, proving to be essential in the dynamics of filopodia and therein in the promotion of migration and invasion of bladder cancer cell [93]. Similar to DARS-AS1, UCA acts as sponge-lncRNA binding itself to miR-145 (Figure 3F). The seed sequence of miR-145 is capable to recognize the 3′UTR of actin-binding protein fascin homologue 1- FSCN1 triggering its mRNA degradation. Downregulation of FSCN1 expression results in impaired and non-functional shorter filopodia around the cell surface [94]. Moreover, miR-145 binds itself to 3′UTR of ZEB1/ZEB2 proteins, known EMT inductors. In line with miR-145 sponging by UCA1, over-expression of it leads to the upregulation of FSCN1 and ZEB1/ZEB2 factors promoting bladder cancer development by induction of EMT process and increasing the invasiveness potential [95,96]. Evenly, knockout UCA1 bladder cell lines displayed a reduction of cellular periphery and diminished the capacity of maligned cells to spread out through other tissues and therein metastasize, suggesting that UCA1 can be used as potential therapeutic target in bladder cancer prognosis [93].

## 4. lncRNAs as Modulators of Rho/ROCK Signaling in Tumorigenesis

Cytoskeletal remodeling is subject to the activity of several receptor proteins on the surface of the cell, which in turn trigger the activation or repression of different intracellular signal cascades involved in cytoskeleton dynamics and turnover of actin dependent structures. Broadly, the RHO GTPases family acts as nexus between signals that come from transmembrane protein receptors and different modulators of cytoskeletal architecture (Figure 4A). Curiously, RHO GTPases dependent intracellular signal cascades regulate the formation of lamellipodia, filopodia, and invadopodia and the release of extracellular matrix metalloproteinases (MMPs) and therein modulate the ability of cells to move and spread out through different tissues [97,98]. Among RHO GTPases family members, three of them have been widely described as pivotal factors in cytoskeletal biology: Rho (RhoA, RhoB, and RhoC), Rac (Rac1, Rac2, and Rac3), and cell division cycle 42 (Cdc42) [99,100]. Rho aggregates actin and myosin to form stress fibers and focal adhesion complex assembly. RhoA and RhoC are present in the cytoplasm, which are activated, while RhoB is found in forming endosomes and at the cellular membrane [101]. RhoA exerts a pivotal function in cytoskeleton derived structures regulating generation of actin-myosin bundles, stress fibers, focal adhesions, and lamellipodia [102,103,104,105,106]. Unlike RhoA, RhoB and RhoC modulate intracellular vesicles formation and endocytosis by participating on endosome and phagosome establishment, respectively, through actin cytoskeletal remodeling [107,108]. Similar to RhoA, Rac members are related to cellular protrusion formation [109,110]. Mainly, Rac1 is a critical factor for the formation of lamellipodia and invadopodia, while Rac3 is required for integrating the adhesion of invadopodia to the extracellular matrix (ECM) to allow it to degrade the ECM [111,112,113]. Rac2 exhibits a structural function modulating the cell adhesion to intercellular adhesion molecule-1 (ICAM-1) [114]. Finally, Cdc42 is an activator of filopodia formation, increasing cell migration and invasiveness potential of malignant cells [115,116,117]. Moreover, Cdc42 is described as critical kinase involved in the regulation of the balance between cellular proliferation and differentiation in several pathological contexts, including carcinomas [35,36]. RHO GTPases members modulate by downstream activation of ROCK proteins. ROCK family includes two members, ROCK1 and ROCK2 [118]. ROCK1 plays a key role in the formation of stress fibers, and it is mainly responsible for rigidity-dependent invadopodia activity through actomyosin contractility [39]. ROCK2 is important for the phagocytosis, cell contraction, and stabilizing of the cytoskeleton [37,38,119]. Intracellular signals mediated by Rho GTPases and ROCK1/2 are responsible of development and balance of formation of lamellipodia, filopodia, and invadopodia, and these signals promote the degradation of the extracellular matrix [39,120]. Underling layer regulation of Rho/ROCK signaling involves several modulators, including non-coding transcripts as microRNAs and/or lncRNAs [121,122]. lncRNAs have demonstrated their importance as effectors of actin cytoskeletal architecture not only by directly interacting with actin or accessory proteins but also by regulating several pathways related to F-actin remodeling such as Rho/ROCK signaling (Figure 3).

Regulation of Rho/ROCK signaling by metastasis-associated lung adenocarcinoma transcript 1 (Malat1) has been described in both osteosarcoma and breast cancer [123,124]. Loss of function assays have demonstrated that the downregulation of Malat1 leads to a similar cancer phenotype, displaying a lower ratio of proliferation and migration and delayed tumor growth accompanied by disruption of actin stress fibers and increasing apoptosis in both tumors. In line with similar phenotype observed in both tumors, Malat1 is highly upregulated in them, suggesting that it promotes tumorigenesis. Curiously, Malat1 exerts oncogene function by different mechanisms depending on cancer type. Cai et al. (2015) [123] showed that Malat1 is required for correct expression of three pivotal factors of Rho/ROCK signaling, RhoA, ROCK1, and ROCK2 in osteosarcoma cell lines and downregulation of Malat1 is translated into disruption of their protein levels. However, the molecular mechanisms responsible for modulation of RhoA, ROCK 1, and ROCK2 by Malat1 remains unknown [123]. By contrast, Malat1 oncogene function in breast cancer resulted in upregulation of Cdc42 by targeting and binding miR-1, which is capable of recognizing the Cdc42 3′UTR and triggering its mRNA degradation. Removal of miR-1 by Malat1 promotes upregulation of Cdc42 protein levels, leading to filopodia formation and therein increased migration and invasiveness of breast maligned cells [124] (Figure 4B).

The molecular function of Rho and Rac members is required for correct actin remodeling in several carcinomas [40,125]. Two lncRNAs, AFAP1-AS1 and SchLAh, have been reported as pivotal regulators of RhoA-Rac1/2 signaling. Bo et al. (2015) [41] demonstrated a pivotal role of Actin filament associated protein 1 antisense RNA1 (AFAP1-AS1) enhancing progression and poor prognoses of nasopharyngeal carcinoma. Downregulation of AFAP1-AS1 is capable of reducing the ability of malignant cells to promote metastasis by triggering loss of integrity of actin filaments accompanied of low rate of cellular migration [41]. Unlike AFAP1-AS1, SchLAH have been described as protective lncRNA in hepatocellular carcinoma. Upregulation of SchLAH is capable of repressing cellular migration and therein metastasis both in vitro and in vivo in lung tissues and HHC cell lines. Likewise, downregulation of SchLAH by siRNAs enhances the ability of HCC cells to migrate. Mechanistically, SchLAH physically interacts with Fused in sarcoma/translocated in liposarcoma (FUS/TLS or FUS), a multifunctional DNA/RNA-binding protein. As result of binding of SchLAH-FUS, several gene pathways are downregulated including failure expression of RhoA and Rac1 which are required to promote cellular invasion potential [126]. Interestingly, the maternally expressed 3 (MEG3) acts as protective lncRNA inhibiting the migration and metastasis of thyroid carcinoma by repression of Rac1 [127] (Figure 4C).

ROCK proteins constitute pivotal effectors downstream of RHO GTPases signals and disruption of their expression leads to failure of RHO GTPases dependent cytoskeleton remodeling [128]. Recently, long non-coding RNA Ewing sarcoma-associated transcript 1 (EWSAT1) has been described as inductor of osteosarcoma (OS) exerting its function like an sponge-lncRNA by binding to miR-24-3p which in turn recognizes ROCK1 3′UTR and represses its translation. EWAST1-miR-24-3p complex results in upregulation of ROCK1 promoting actin stress fiber formation and migration. Furthermore, EWSAT1 expression is upregulated in OS and increased EWSAT1 expression is correlated with poor prognosis in OS patients [129] (Figure 4D).

## 5. Conclusions and Future Perspectives

The analysis of the human transcriptome has made possible to elucidate the importance of the non-coding genome in the regulation of multiple diseases such as cancer. Interestingly, tumorigenesis requires remodeling of the cytoskeleton, and thereafter, malignant cells can migrate and disperse through the organism, colonizing other tissues in a process term metastasis, which promotes the aggressiveness of cancer and therefore is correlated with a poor diagnosis of disease. The regulatory potential of lncRNAs has been described in all cellular processes including the modulation of the cytoskeletal architecture by regulating both structural cytoskeletal proteins and the signaling pathways that trigger the formation of actin-dependent structures. Several reports have highlighted the importance of lncRNAs as oncogenes in multiple carcinomas suggesting their pivotal role in the necessary remodeling of the cytoskeleton that leads to metastasis. Furthermore, knockout of most lncRNAs involved in cytoskeletal homeostasis leads to loss of ability to spread out and lower proliferation and survival rate pinpointing that lncRNAs can be considered as potential therapeutic targets against several carcinomas. Curiously, just a few cases of lncRNAs have been reported as regulators of cell–cell interactions in other contexts such as atrial fibrillation or even in non-pathological processes, demonstrating the importance of lncRNAs in cytoskeletal homeostasis. Given the relevance of cytoskeletal lncRNAs as modulators of migration and cell cycle regulation described above, it would be interesting to analyze their role during embryonic development since this complex process requires a tight and precise migration of different cell subpopulations. Failure of proper cell migration during embryonic development would result in malformation of the embryo and thus in many cases in congenital diseases that ultimately might lead to embryonic death.

## Figures and Tables

**Figure 1 ncrna-08-00028-f001:**
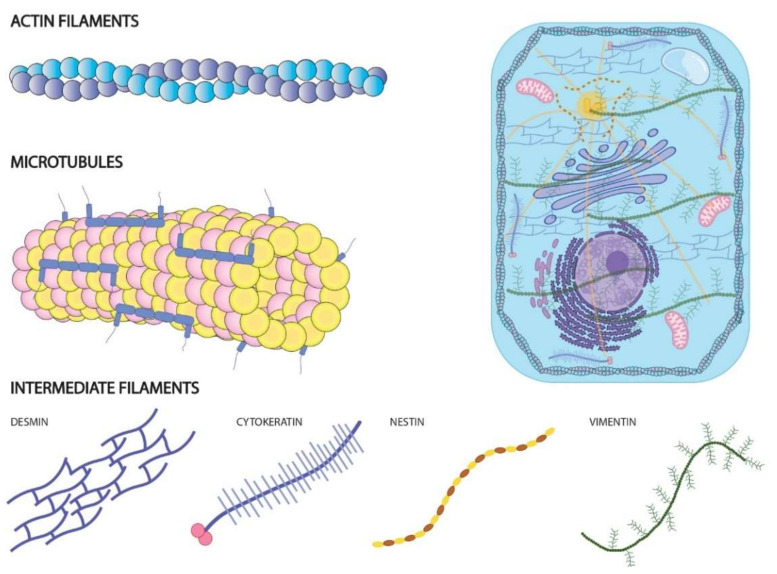
Schematic representation of three main cytoskeletal protein structures. Actin filaments are responsible for shape and movement of cells; Microtubules’ function is essential for cell cycle progression; intermediate filaments exert a structural function in cytoskeletal architecture.

**Figure 2 ncrna-08-00028-f002:**
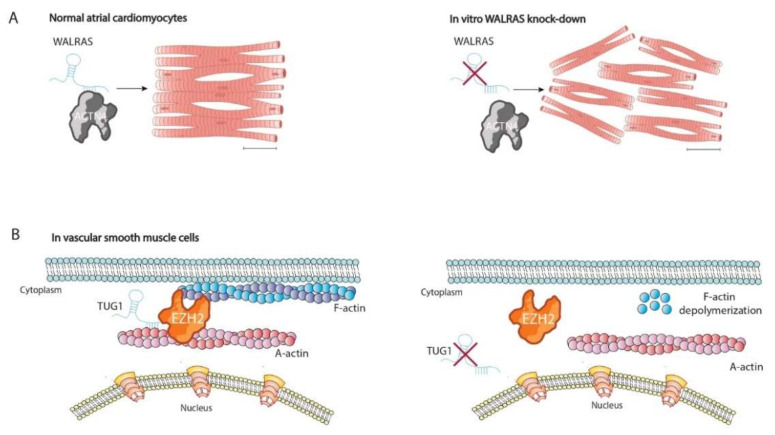
Schematic representation of cytoskeletal lncRNAs involved in cellular homeostasis. (**A**) Walras is required for the integrity of actin fibers in cardiomyocytes. (**B**) TUG1 modulates methylation of α-actin by EZH2 and therein is required for the formation of the cortex cytoskeleton.

**Figure 3 ncrna-08-00028-f003:**
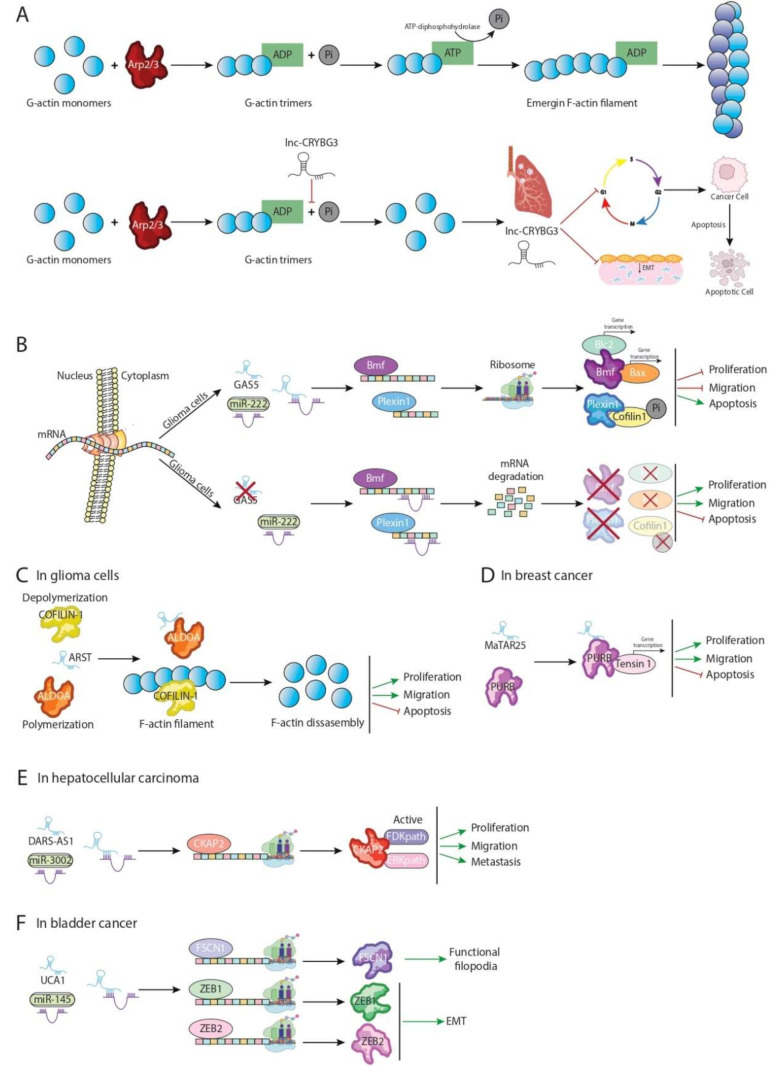
Schematic representation of several mechanisms related to cytoskeletal lncRNAs function: (**A**) blocking phosphorylation of G-actin mediated by Lnc-CRYBG3; (**B**) sponge GAS5-miR 222 enhances Bmf and Plexin1 translation leading to a reduced cellular proliferation and migration ratio; (**C**) ARST binds to ALDOA avoiding its interaction with F-actin, and in turn, Cofilin-1 binds to F-actin filament increasing the disassembly ratio; (**D**) MaTaR25-PURB complex is required for correct expression of Tensin1, which in turn increases proliferation and migration of glioma cells; (**E**) DARS-AS1 increases expression of CKAP2, which in turn represses FAK and ERK expression; (**F**) sponge UCA1-miR-145 enhances FSCN1, ZEB1/ZEB2 translation leading to filopodia formation and increasing EMT process.

**Figure 4 ncrna-08-00028-f004:**
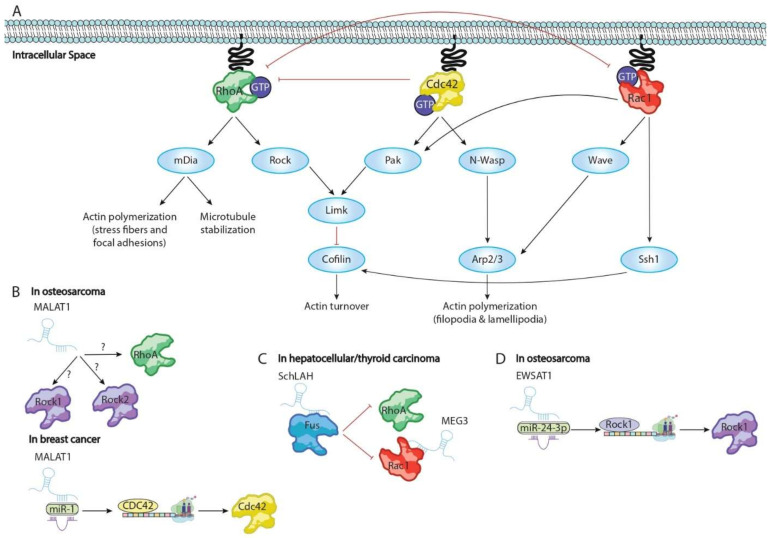
Schematic representation of lncRNAs involved in Rho/ROCK signaling: (**A**) pathways mediated by Rho/ROCK signaling; (**B**) Malat1 increases RhoA, ROCK1, and ROCK2 expression in osteosarcoma progression while acting as sponge by binding to miR-1, leading to Cdc42 translation in breast cancer; (**C**) SchLAH represses RhoA and Rac1 interacting with Fus; (**D**) EWSAT1 enhances ROCK1 translation by avoiding that miR-23-3p recognizes ROCK1 3′UTR.

**Table 1 ncrna-08-00028-t001:** Summary of lncRNAs related to cytoskeleton architecture and their impact on tumorigenesis as well as on other diseases.

Cytoskeletal-lncRNAs Related to Tumorigenesis				
**lncRNA**	**Target Molecule**	**Function**	**Tissue or Cell Line**	**Reference**
**LNC-CRYBG3**	G-actin	Inhibition of F-actin polymerization avoiding G-actin phosphorylation	Lung cancer	[28]
**CYTOR**	Golph3, Rhobtb3 and PKP4	Cytoskeletal homeostasis and cell cycle progression	Breast cancer cell line	[29]
**Gas5**	miR-222	Enhance Bmf and PLXN1 expression reducing aggressiveness tumour	U87 and U251 glioma cell line	[30]
**ARST**	ALDOA	Mediate actin fibers integrity avoiding that ALDOA can attach to F-actin binding sites increasing F-actin depolymeration	U87 and U251 glioma cell line	[31]
**MaTaR25**	PURB and Tensin1	Enhance PURB dependent genes remodelling cytoskeleton architecture and increasing migration and spread out of maligned cells	Breast cancer cell line	[32]
**DARS-AS1**	miR-3002	Sponge miR-3002 enhancing CKAP2 translation and aggravating the growth and metastasis of tumor	Hepatocellular carcinoma	[33]
**UCA1**	ZEB1/2 and FSCN1	Increase formation of actin-dependent cell filopodia enhancing metastasis	Bladder carcinoma	[34]
**Malat1**	RhoA, ROCK1 and ROCK2	Increasing RhoA, ROCK1, and ROCK2 translation required to migration and cytoskeletal homeostasis	Osteosarcoma	[35]
**Malat1**	miR-1	Sponge miR-1 enhancing Cdc42 translation required to migration and cytoskeletal homeostasis	Breast carcinoma	[36]
**AFAP1-AS1**	RhoA and Rac1	Enhancing progression and poor prognosis of nasopharyngeal carcinoma increasing capacity of spreading out	nasopharyngeal carcinoma	[37]
**SchLAH**	FUS/TLS	Repressing cellular migration and therein metastasis triggering downregulation of RhoA/Rac2 signalling	Lung carcinoma	[38]
**EWAST1**	miR-24-3p	Sponge miR-24-3p enhancing expression of ROCK1 and promoting actin stress fiber formation and migration	Osteosarcoma	[39]
**Walras**	ACTN4	Required to actin cytoskeleton integrity	Cardiomyocites	[40]
**TUG1**	EZH2 and actin	Methylation of α-actin by EZH2	Vascular smooth muscle cells	[41]

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
