# Peer review of "New Insights into the Roles of lncRNAs as Modulators of Cytoskeleton Architecture and Their Implications in Cellular Homeostasis and in Tumorigenesis"

_ncrna, 2022, doi:10.3390/ncrna8020028_

Round 1

Reviewer 1 Report

Authors have replied all the concerns, thus I reccomend to accept the manuscript for publication in its actual revised form.

Author Response

Thank you very much for your comments, which have undoubtedly given considerable notice to the quality of the review.

Reviewer 2 Report

Garcia-Padilla et al., revised their manuscript by adding new Figures and a new Table. These modifications succeeded to enhance readability of the text.  Still, the manuscript has several concerns. The comments below may help to further improve the manuscript.

Major comments:

1.   Line 86, 322, 324, 327, 345, 395, 396, 395, and 396.

      “RhoGTP”………Rho GTPases mean the Rho family of small GTPases and include Rho, Rac, and Cdc42. The Rho protein takes two forms: Rho-GTP and Rho-GDP bound forms. Please do not confuse Rho GTPase from RhoGTP.

2.   Table 1.

       LNC-CRYB63   → LNC-CRYBG3

      cycle cell progression  → cell cycle progression

       despolymeration  → depolymerization

       Osteosarcome   → Osteosarcoma

3.   The authors are recommended to clearly put the figure number in the corresponding text.

      For example:  

Line 148.   G-actin preventing it from polymerizing into F-actin (Fig. 2A).

Line 216.   Gas5 repressed … as an sponge lncRNA of it (Figure 2B).

Line 236.   But modulates functional binding of Cofilin-1 to F-actin (Figure 2C).

Line 268.   in cytoskeleton architecture [67] (Figure 2D).

Line 295.   and cellular proliferation and migration [72] (Figure 2E).

Line 307.   lncRNA binding to miR-145 (Figure 2F).

        etc.

Minor comments: 

1.   Figures. Please remove the old figures.

2.   Line 100.  the role lncRNAs  →  the role of lncRNAs

4.   Line 332.  RhoA exert → RhoA exerts

5.  Line 340.  Rac2 exhibit → Rac2 exhibits

6.  Line 398.   (SO) → (OS)

8.  Line, 412.  LncRNAs cytoskeletal modulators of homeostasis cellular

                        “LncRNAs as cytoskeletal modulators of cellular homeostasis” or

                       “LncRNAs modulate cellular cytoskeletal homeostasis”

9.  Line 430.    EZH2 mediate → EZH2 mediated?

Author Response

First of all, we would like to thank the reviewer for their constructive comments that certainly will be helpful in this manuscript.

 Major comments:

  1. Line 86, 322, 324, 327, 345, 395, 396, 395, and 396.

      “RhoGTP”………Rho GTPases mean the Rho family of small GTPases and include Rho, Rac, and Cdc42. The Rho protein takes two forms: Rho-GTP and Rho-GDP bound forms. Please do not confuse Rho GTPase from RhoGTP. Clearly, we were confused with these terms and we have changed them in the text to avoid the confusion of the readers.

  1. Table 1.

       LNC-CRYB63   → LNC-CRYBG3 Changed added

      cycle cell progression  → cell cycle progression Changed added

       despolymeration  → depolymerisation Changed added

       Osteosarcome   → Osteosarcoma Changed added

  1. The authors are recommended to clearly put the figure number in the corresponding text.

Following the reviewer´s recommendation, we have included references of figures into the text. 

      For example:  

Line 148.   G-actin preventing it from polymerizing into F-actin (Fig. 2A). Changed added

Line 216.   Gas5 repressed … as an sponge lncRNA of it (Figure 2B). Changed added

Line 236.   But modulates functional binding of Cofilin-1 to F-actin (Figure 2C). Changed added

Line 268.   in cytoskeleton architecture [67] (Figure 2D). Changed added

Line 295.   and cellular proliferation and migration [72] (Figure 2E). Changed added

Line 307.   lncRNA binding to miR-145 (Figure 2F). Changed added

Reviewer 3 Report

García-Padilla et al. summarize information from the literature about lncRNAs implicated in cell architecture, and highlight the importance of cell integrity and adhesion in diseases such as cancer and muscular disorders. The subject is interesting and the manuscript is beautifully illustrated. The text is well structured and informative; however, it needs to be thoroughly revised by a native English speaker because it is riddled with typographical and grammatical errors and numerous spelling mistakes.

A few examples illustrate what I mean:

Title: New insights into the roles of lncRNAs as…

Line 17: The importance…

L19: …, the cytoskeleton…

L22: that sentence needs to be rewritten: “…, which have a preferably structural function showing specificity for the cell type in which are expressed”.

L24: The cytoskeleton…

L29: …about the roles of…

L46: replace “suffer” by “undergo”

L46: Evolutionary analyses have demonstrated…

L56: replace “reflexing” by “reflecting”

L86: the sentence “post-transcriptional has been described exerting modulation of several genes of these pathways” makes no sense.

L97: replace “manuscript” by “review”

L206: replace “exerting as an sponge lncRNA of it” by “acting as a competitive endogenous lncRNA (ceRNA)”

L207: replace “inductor” by “inducer”

L300: “trigger activation and / or inactivation” does not make a whole lot of sense.

L340: the sentence “downregulation of Malat1 results in similar cancer phenotype loosing of both tumors displaying a lower ratio of proliferation and migration and delayed tumor growth accompanied to disruption of actin stress fibers and increasing apoptosis cellular” (like many others in the text) appears to have been translated by an algorithm (Google translate?) and needs to be rewritten.

L389: should read “lncRNAs are cytoskeletal modulators of cellular homeostasis”

L419: should read “Conclusions and future perspectives”

Author Response

First of all, we would like to thank the reviewer for their constructive comments that certainly will be helpful in this manuscript.

García-Padilla et al. summarize information from the literature about lncRNAs implicated in cell architecture, and highlight the importance of cell integrity and adhesion in diseases such as cancer and muscular disorders. The subject is interesting and the manuscript is beautifully illustrated. The text is well structured and informative; however, it needs to be thoroughly revised by a native English speaker because it is riddled with typographical and grammatical errors and numerous spelling mistakes.

Following the reviewer´s recommendation and to correct typographical and grammatical errors and numerous spelling mistakes the manuscript has been revised by native English speaker

A few examples illustrate what I mean:

Title: New insights into the roles of lncRNAs as… Changed added Changed added

Line 17: The importance… Changed added

L19: …, the cytoskeleton… Changed added

L22: that sentence needs to be rewritten: “…, which have a preferably structural function showing specificity for the cell type in which are expressed”. Changed added

L24: The cytoskeleton… Changed added

L29: …about the roles of… Changed added

L46: replace “suffer” by “undergo” Changed added

L46: Evolutionary analyses have demonstrated… Changed added

L56: replace “reflexing” by “reflecting” Changed added

L86: the sentence “post-transcriptional has been described exerting modulation of several genes of these pathways” makes no sense. Changed added

L97: replace “manuscript” by “review” Changed added

L206: replace “exerting as an sponge lncRNA of it” by “acting as a competitive endogenous lncRNA (ceRNA)” Changed added

L207: replace “inductor” by “inducer”

L300: “trigger activation and / or inactivation” does not make a whole lot of sense. Changed added

L340: the sentence “downregulation of Malat1 results in similar cancer phenotype loosing of both tumors displaying a lower ratio of proliferation and migration and delayed tumor growth accompanied to disruption of actin stress fibers and increasing apoptosis cellular” (like many others in the text) appears to have been translated by an algorithm (Google translate?) and needs to be rewritten. Sentence has been changed

L389: should read “lncRNAs are cytoskeletal modulators of cellular homeostasis” Changed added

L419: should read “Conclusions and future perspectives” Changed added

Reviewer 4 Report

The manuscript entitled “New Insights of lncRNAs as Modulators of Cytoskeleton Architecture and their Implications in Cellular Homeostasis” authored by Garcìa-Padilla represents an interesting Review talking about the implicated molecular mechanisms in modulating the cellular cytoskeleton architecture during homeostasis and tumorigenesis. The role of different structural proteins and their functions were well described as well as the changes occurred during tumorigenesis. The implicated mechanisms were then discussed with representative figures that allow a better understanding of the explained information. The manuscript is well written. However, few minor points should be addressed in order to render the manuscript suitable for publication.

Minor points to be addressed

  • The title should be improved to highlight the role of lncRNAs also in tumorigenesis.
  • In the introduction section, there is evidenced a lack of the connection between the cytoskeletal homeostasis and pathology before passing through the tumorigenesis. It is suggested to better clarify in this section how cytoskeletal homeostasis is disrupted and how this disruption might be linked to tumorigenesis.
  • The table 1 concerning the “lncRNAs related to cytoskeleton architecture and their impact in tumorigenesis” should be introduced as separate paragraph (lines 88 – 103) to emphasize the role of lncRNAs in tumorigenesis. Please improved it.
  • Line 275, please adjust the term “FAK” and not “FDK”.
  • The section 4 “LncRNAs cytoskeletal modulators of homeostasis cellular” can be moved after the Introduction section in which the authors might start talking about the implication of lncRNAs cytoskeletal modulators of homeostasis cellular in general and then go to deepen the role of lncRNAs in tumorigenesis with the well-developed implicated pathways.

Author Response

First of all, we would like to thank the reviewer for their constructive comments that certainly will be helpful in this manuscript.

The manuscript entitled “New Insights of lncRNAs as Modulators of Cytoskeleton Architecture and their Implications in Cellular Homeostasis” authored by Garcìa-Padilla represents an interesting Review talking about the implicated molecular mechanisms in modulating the cellular cytoskeleton architecture during homeostasis and tumorigenesis. The role of different structural proteins and their functions were well described as well as the changes occurred during tumorigenesis. The implicated mechanisms were then discussed with representative figures that allow a better understanding of the explained information. The manuscript is well written. However, few minor points should be addressed in order to render the manuscript suitable for publication.

Minor points to be addressed

The title should be improved to highlight the role of lncRNAs also in tumorigenesis. Following the reviewer´s recommendation the title has been changed: New Insights into the roles of lncRNAs as Modulators of Cytoskeleton Architecture and their Implications in Cellular Homeostasis and in Tumorigenesis

In the introduction section, there is evidenced a lack of the connection between the cytoskeletal homeostasis and pathology before passing through the tumorigenesis. It is suggested to better clarify in this section how cytoskeletal homeostasis is disrupted and how this disruption might be linked to tumorigenesis. Following the reviewer´s recommendation we have connected better cytoskeletal homeostasis and pathology relationship

  • The table 1 concerning the “lncRNAs related to cytoskeleton architecture and their impact in tumorigenesis” should be introduced as separate paragraph (lines 88 – 103) to emphasize the role of lncRNAs in tumorigenesis. Please improved it. Following the reviewer´s recommendation table 1 have been introduced as separate paragraph

  • Line 275, please adjust the term “FAK” and not “FDK”. Changed added
  • The section 4 “LncRNAs cytoskeletal modulators of homeostasis cellular” can be moved after the Introduction section in which the authors might start talking about the implication of lncRNAs cytoskeletal modulators of homeostasis cellular in general and then go to deepen the role of lncRNAs in tumorigenesis with the well-developed implicated pathways. Following the reviewer´s recommendation section 4 has been changed to section 2

Round 2

Reviewer 2 Report

Garcia-Padilla et al., revised their manuscript in response to my comments.  Still, the manuscript has several concerns. The comments below may help to further improve the manuscript.

Major comments:

1. It may be necessary for you to read this manuscript as you read the reviews written by other researchers.

Minor comments: 

1.   Line 86, 357, 359, 363, 381, 431, and 432, .    RHOGTPases   → Rho GTPases 

2.    Line 89.   delete “into”.

3.   Line 92.  shape cellular  → cell shape

4.   Line 92.   process → processes

5.  Line 93.   Several  → several

6.  Line 97, 99, 124, 400, and Table 1. 

  You used “tumour” and “tumor”. Which one do you want to use?  

7.  Line 105.   display → displays

8.  Line, 106.  are-correlated    are correlated 

9.  Line 133.   lost    loss

10.   Line 157.   microfilament    microfilaments

11.   Line 199.  the cytokinesis ring from forming    the formation of contractile ring during   cytokinesis

12.  Line 298.  dynamics stability of FAs    “dynamics of FAs” or “stability of FAs”

13.  Line 336.  determinate  → determines

14.  Line 399.  accompanied to  → accompanied by

15.  Line 400.  apoptosis cellular    apoptosis

16.  Line 403.   depending of    depending on 

17.   Line 419.   accompanied to   → accompanied by 

18.  Line 424.  ability migration     the ability of migration

This manuscript is a resubmission of an earlier submission. The following is a list of the peer review reports and author responses from that submission.

Round 1

Reviewer 1 Report

Garcia-Padilla at al., summarized the recent publications on the role of lncRNAs on actin cytoskeleton. The selection of papers appears to be comprehensive and appropriate. But I felt difficulty to read this manuscript. I hope that my comments below may be helpful to improve the manuscript.

Major comments:

1.   Figure 1.

       Readers cannot assign the location of each cytoskeleton in a cell. It is recommended to use the color code for assignment. Four types of intermediate filaments needs their names.  

2.   Figure 2.

       It is better to show the binding of ATP, ADP, and ADP + Pi for all actins in this figure. Otherwise, it is difficult to see which step is inhibited by LNC CRYG63.

3.   Figure 3.

       Does GAS5 bind 4 molecules of miR-222? What do Burgundy and Green substances mean? Please add the succinct and complete legend.

4.   Because this review deals with so many lncRNAs with different functions, I felt difficulty in reading the text. Please add a new Table that lists lncRNAs with a target molecule, assumed function of each lncRNA, the name of tissue or cells in which lncRNA exists, and the number of reference. 

Minor comments: 

1.   Line 76.  Squematic  → Schematic

2.   Line 92.  establishing proteins … I cannot understand the meaning. Please reconsider this phrase.

3.   Line 95.  is-correlated   → is correlated  

4.   Line 102:  LncRNAs modulators  → LncRNAs modulate?

5.  Line 118.   Polymeration actin  → Polymerization of G-actin?

6.  Line 137.   Pei et al., (2018),   → Pei et al., (2018) [33],

8.  Line, 138, 140, 141, 143, 144, 148, 150, 155, 158.

                        CRYB63     CRYBG3

9.  Line 141.    is binding directly  → binds directly

10.  Line 159.   Grembergem et al., (2016)  → Van Grembergen et al., (2016) [36]

11.  Line 192.    Zhao et al. (2015)   → Zhao et al. (2015) [49]

12.  Line 197, 198, 207.     Plexin 1  → Plexin C1

13.  Line 239.  Chang et al., (2021)  →  Chang et al., (2021) [63]

14.   Line 301.  Extracellular   → extracellular

15.  Line 317.  promotor  …. Because this word belongs to the technical term in genetics, please use another word.

Reviewer 2 Report

Reviewer Comments:

García-Padilla, C and coworker present the manuscript entitled “New Insights of lncRNAs as Modulators of Cytoskeleton Architecture and their Implications in Cellular Homeostasis”. The authors presented an interesting and meaningful research. However, I have some concerns that need to be addressed before potential publication of manuscript.

Minor comments:

Abstract:

Page 1, line 19. “transduction” can be corrected to “the transduction”.

Page 1, line 21. “mantaining” can be corrected to “maintaining”.

Page 1, line 29 “network” can be corrected to “the network”.

Page 1, line 31 “in” can be corrected to “on”.

Page 1, line 32 “not related” can be corrected to “unrelated”.

Introduction

The authors should rewrite this part "deep advances in transcriptomic and genomics have led to elucidate that the genome is far more pervasively transcribed than was previously appreciated" is not understood.

Are lncRNAs located in both cellular compartments (nucleus and cytoplasm) transcriptional or post-transcriptional regulators? Why are they located in both compartments?

Page 3, line 82, Delete the space before the punctuation symbol in “levels,”

Page 3, line 94 “most of cytoskeletal” can be corrected to “most cytoskeletal”.

Page 3, line 96. “manuscript we” can be corrected to “manuscript, we”.

Page 3, line 100 “herein” can be corrected to “here in”.

Page 3, line 111 “of” can be corrected to “from”.

Abbreviations should be defined at their first appearance. However, some abbreviations were treated backwards, such as Cofilin-1 (CFL-1) on line 179 on page 5 and line 129 on page 3. Also, some abbreviations were not defined, such as FMNL1 on line 128 page 3. Please check all these details.  

The authors should redesign the figures, they are very simple and do not describe the molecular mechanisms and biological processes ullustrated in the desings, for example the lncRNA CRYB63 induces apoptosis and inhibits proliferation and migration in lung cancer cell lines, this is not described in the figure.  

Figures 2 and 3 are not mentioned in the text. The authors should review these details.

Figures 2 and 3 will be merged into one and redesigned to include all the described lncRNAs that function as modulators of actin filaments and accessory proteins in cancer pathogenesis and the biological processes they regulate.

Page 5, line 196 “expression” can be corrected to “the expression”.

I consider it important to include a figure of LncRNAs as cytoskeletal modulators of cellular homeostasis, as this is the focus of this review.

Page 9, line 379 “modulators” can be corrected to “modulator”.

Page 9, line 396 “assay” can be corrected to “assays”.

Page 9, line 404 “a process terms” can be corrected to “a process term”

Page 9, line 418 Replace ' such' with ', such'.

Page 9, line 419, include space before Since.

Finally, two new figures illustrating the data presented in the paragraphs “3. LncRNAs as modulators of Rho/ROCK signaling in tumorigenesis”, and “4. LncRNAs cytoskeletal modulators of homeostasis cellular” must be very informative and useful.